# The Use of Gamification and Web-Based Apps for Sustainability Education

Carolina Novo [1], Chiara Zanchetta [1], Elisa Goldmann [2] and Carlos Vaz de Carvalho [3,*]

1    Virtual Campus, 4350-151 Porto, Portugal; carolina_novo@virtual-campus.eu (C.N.);
chiara_zanchetta@virtual-campus.eu (C.Z.)
2    Center for Sustainable Governance (CSG), Fachhochschule des Mittelstands (FHM), 33602 Bielefeld, Germany;
elisa.goldmann@fh-mittelstand.de
3    Games, Interaction and Learning Techonologies R&D, Instituto Superior de Engenharia do Porto,
4200-072 Porto, Portugal
*    Correspondence: cmc@isep.ipp.pt

**Abstract:** This article dwells on the role of gamified digital tools in promoting environmental self-awareness and action. In particular, it unfolds the outreach of a web application, developed within the European GoBeEco project, aimed at encouraging users to adopt ecological and sustainability habits. In this article, the focus is on the implementation of the project in Portugal, and, therefore, the data presented in the results reflect the involvement of participants on a national level. Overall, more than two dozen participants were involved in the validation of the application, which comprised three evaluation phases involving the distribution of questionnaires and the organisation of a focus group aimed at assessing the role of GoBeEco in fostering sustainable personal change and also evaluating specifically the role that gamified elements played in that change. Results show that the application had a very positive impact on the users and helped mitigate the well-documented gap between sustainable awareness and action, and, also, that the gamification strategy contributed to that purpose. We argue that the potential of these applications in Portugal is latent but still has room for growth. In this sense, the study also highlights future paths for the development and implementation of these tools, based on the features most valued by users—access to statistical data, examples from real life, gamified and fun elements, and focus on daily, individual actions, among others.

**Keywords:** gamification; sustainability; education; web-app; behavioural influence; ecology

## 1. Introduction

In June 1972, the United Nations held its inaugural conference on international environmental issues in Stockholm, marking a significant and pivotal point in the evolution of environmental policies. This event underscored the importance of pollution as a major economic issue and elevated the pursuit of a sustainable environment to a high priority on the worldwide agenda. In the subsequent years, numerous initiatives were undertaken both internationally and nationally, utilizing political and legal tools, to address and mitigate what became increasingly recognized as a paramount concern: the environmental crisis [1].

From that moment, it also stood as equally relevant to capacitating citizens to understand climate issues and to act accordingly. In this matter, it can be said that citizens are, in the present time, relatively well informed about the impact of individual actions on environment, and defend a consorted action between political institutions, private companies, industries, and citizens in general. The most recent Special Eurobarometer survey regarding the attitudes of the European citizens about the environment, launched in March 2020, concluded that Europeans have been presenting some behavioural changes and adopting a more conscious perspective towards the environment: for example, "[...] two-thirds [...] say that they have separated most of their waste for recycling, while the next most common activities are avoiding single-use plastic goods or buying reusable

plastic products (45%), buying local products (42%) and cutting down energy consumption (37%)" [2].

Nevertheless, some of the participants admitted being aware that they do not do enough and that this eventually will have consequences on their daily routine and health. While the great majority of respondents shared that it is important to "change the way we consume" and that it is necessary to "change the way we produce and trade", many also highlighted that one of the most effective ways of promoting it is to provide "[...] more information and education" [2]. Another study, "Attitudes of European citizens towards the environment", a special Eurobarometer Report, corroborates this point, by stating that "(...) consumers differ in terms of their environmental consciousness and behaviours" [3]. This reflects what some empirical studies showed: despite a heightened level of environmental awareness, encompassing both emotional and cognitive aspects, there appears to be a significant disparity between this awareness and actual behaviour so, while there is a widespread belief that society has reached its limits and change is demanded, the manifestation of environmentally friendly behaviours do not accompany that belief [4]. Thus, the gap between environmental attitudes and actions remains a notable challenge. In the same reference, Kollmuss and Agyeman underscored the role that entrenched behavioural patterns play as a potent but often overlooked obstacle in the literature on pro-environmental behaviour [4]. Another viewpoint is that many individuals do not perceive the environment as a shared resource, even if acknowledging that "one individual's consumption of natural resources also affects other people" [5].

Consequently, recognising that individual actions collectively impact the environment is pivotal. Despite scepticism about the impact of single actions, the cumulative effect can be substantial. This perspective aligns ecological behaviour with an altruistic stance, characterised by considerations that improve others' situations at one's own expense [5]. In addition to altruistic thinking, environmental awareness emerges as another influential factor, which can be instilled through environmental education strategies. These educational initiatives aim to cultivate a heightened consciousness about environmental issues and promote ecologically responsible actions among the next generation of professionals [6]. In this sense, it becomes clear that increasing the variety and availability of information and tools has the potential to increase citizens' knowledge and therefore incite informed and effective actions. It is also important that people do not feel discouraged or impotent in a world of massive and overwhelming chances and understand that their individual actions matter. They should be encouraged to do so, by knowing that, for example, the transformation of individual habits sends a clear message to industries and governments that environmental preservation is a shared priority and that ultimately, it is through the cumulative impact of these altered habits that we can pave the way for a greener and more sustainable future for generations to come. Similarly, by focusing on changing habits in a consistent, gradual and informed way, education strategies focused on changing behaviours might help individuals overcome the aforementioned "old habits" which seem to be so pervasive, and, hopefully, change, day by day, what they do, until the new behaviour becomes consistent and natural enough to replace the previous one.

The project under review in this article constitutes, hopefully, a small but significant step in that direction. GoBeEco, an initiative co-funded by the European Union and implemented from 2020 to 2023 in the field of adult education, was coordinated by the Fachhochschule des Mittelstands (FHM), in Germany, and counted with the collaboration of Energie Impuls OWL (Germany), PAIZ Konsulting and Ekopotencjal Foundation (Poland) and Virtual Campus (Portugal). It arose in the context of a well-researched gap between environmental consciousness and action, and to promote sustainable action and pro-ecological behaviours in daily life activities, providing citizens with comprehensive information on the topics and assisting them in transitioning to a greener and more sustainable way of living. The project aimed at contributing to the achievement of some Sustainable Development Goals (SDGs) in the partner countries. Firstly, we can state that the GoBeEco project, and its results, to be explored throughout this article, constituted a step forward in achieving

Goal 4, Quality Education, as the project—through its main results, the gamified web-app and the Digital Edu Skills Handbook (available at https://www.gobeeco.eu/, accessed on 08 April 2024) targeting educators—enhanced environmental education strategies to cultivate a heightened consciousness about environmental issues among adults, thereby promoting lifelong learning opportunities. The Handbook, in particular, is believed to contribute to this goal, as it offered educators curricula and methodologies that could be used as a benchmark for a wide selection of educational goals and audiences, as it provided them with a learning roadmap to structure their classes and with a set of digital learning resources. It is believed that educators would be able to apply the educational process learnt in the Handbook when working with all available applications for mobile devices that appear on the market. Moreover, the Handbook also contains a library of examples, ideas, best practices and digital tools available online for developing pro-ecological habits. It also has a section with tests for educators to check their knowledge of innovative teaching methods and digital tools, as well as methods for assessing their students' skills and knowledge. Through the application and the Handbook, the project also tried to answer a perceived need, stated by the educators involved in the project (and expanded in the needs analysis, in Section 2.1 of this article) of having more information and resources available and the interest of having an application with sustainable challenges at their disposal.

Secondly, and more centrally, the project is perfectly aligned with Goal 12 (Responsible Consumption and Production), through its emphasis on promoting sustainable actions and behaviours, GoBeEco supports efforts to achieve sustainable consumption and production patterns, encouraging individuals to make informed choices and reduce waste, choosing better the products and services they purchase, but also capacitating them to act in an alternative way, by sharing information about the impact their actions might have and suggesting other modes of action. Through the missions, challenges and tasks designed for the application (see Section 2.4), it was believed that users were encouraged to adopt better habits in all the dimensions of their lives; to this purpose also contributed the wide range of applications, informative material and interactive platforms and tools available in the Handbook, chosen particularly to help users achieve, in their own lives, the SDGs, as the several resources listed were precisely distributed by the sustainable development goal they aimed at helping to achieve. More indirectly, the project and its outcomes are also contributing to SDG 13, Climate Action, as GoBeEco plays a crucial role in raising awareness about climate issues and facilitating actions that contribute to mitigating climate change, aligning with global efforts to address this pressing challenge, and with Goal 11, Sustainable Cities and Communities, by encompassing several tasks directed at fostering a community-oriented behaviour and supporting learners in putting into practice actions that could contribute to overall better places and spaces (here, the activities related to the public space, workplaces and mobility are of particular relevance). With this paper, we hope not only to draw attention to this project which is believed to constitute a valuable and fairly innovative approach to education for sustainability using applications and digital means (as it is further explained in Section 2.1), but also to contribute to minimising a gap in the research about the use and impact of educational applications to promote ecological behaviour in Portugal, a field that remains quite underexplored. It should be noted that studies have been published regarding the status of education for sustainability in Portugal, as is the case of the research conducted by Torres et al., which aimed to evaluate the implementation of ESD (education for sustainable development) in a Portuguese public university by analysing the online descriptions of course units (CU) and identifying explicit references to ESD and highlighted the need for an educational program for teachers to be developed and implemented in the future [7]. In a closer path to this article, one book chapter published by Liberato uncovers the benefits of gamification associated with storytelling to promote and enhance the sustainable tourism supply in a particular Portuguese region (Peso da Régua) [8], and Björk and Zandin's [9] paper dwelled on how the use of a digital platform, City Points Cascais, could encourage good citizenship and ecological practices in this Portuguese municipality and came up with good results that reinforced the gamification potential in

effectively engaging citizens and encouraging sustainable behaviours. However, apart from these contributions, a void regarding the specific reality in Portugal was found, which GoBeEco, and the subsequent study, aim at helping to reduce.

*1.1. The Digital World in Favour of Sustainable Habits*

In this context, it becomes particularly important to understand the status quo of education regarding this topic and how educational tools can inspire and teach people about the importance of prioritizing environmental issues, especially when it comes to adult learners. Education and awareness stimulation play pivotal roles in fostering sustainable habits as they can speed up the process of behavioural change, particularly when the methods adopted reflect the preferences of modern learners who like to learn through dynamic, self-directed, online learning opportunities that allow for continuous development. The integration of sustainability-focused topics into the school curriculum has the potential to shape mindsets, instil eco-conscious values, and empower individuals to make informed decisions.

It can be stated with some certainty that the first step in sustainable education—and here we consider education in a long-term, continuous way, believing that it should transpose the classroom setting and actually be transferred into real-life contexts—is raising awareness and facilitating the acquisition of knowledge about these topics. Here, digital tools also come in handy and have been shown to be much more effective in preparing learners intellectually, as it has been as highlighted by several authors. Chin et al. conducted an experiment in schools in Malaysia, incorporating a digital learning platform as the main learning tool in a course aimed at preparing teachers to teach sustainability-related subjects, and to help students gain the knowledge and skills in the management of the environment which can be applied in school as well as daily lives. It was a successful experience, as they came to conclude that the use of digital tools as teaching and learning strategies in the context of sustainable development could be proven to be very effective and have significant advantages when compared with more traditional or even other methods and tools, as this methodology aligns with a transformative approach, which "[...] is able to raise awareness, provide the knowledge and skills for lasting behavioural change in lifestyle and active participation in social issues [...]" [10]. To the authors, the key to success is in the deployment of active learning methodologies, and, in particular, in the combination of the use of games, the implementation of projects, and the engagement through an online platform, as this allows the students to develop informed engagement, feel more empowered and take charge of their learning process, which can, in turn, lead to a long-lasting knowledge and change of beliefs [10]. All this is carried out while respecting and valuing learners' expectations: learning at their own pace, using digital applications, learning through dynamic, self-directed and continuous opportunities [11,12].

The deployment of digital resources in education, to foster active learning, can happen, for example, through the introduction of gamification. Gamification is the process of using game elements (principles, mechanics and others) in non-game contexts to increase the motivation and engagement of the users or participants so that they are able to reach their (intrinsic or extrinsic) goals [13,14]. Gamified learning involves using game-based elements such as point scoring, peer competition, teamwork, score tables to drive engagement, help students assimilate new information and test their knowledge [15]. It can be applied to school-based subjects, but it is also used widely in self-teaching applications and courses. The most effective gamification systems in education seem to make use of elements such as narrative and connection with fellow players/learners to really capture the learner's interest. Examples of gamified elements that can be used in education are the following [16]:

- Points when achieving academic objectives (where correct or well-structured answers operate on a points system, with students moving up through the ranks).
- Points for meeting procedural/non-academic objectives such as helping colleagues, tidying the classroom, etc.
- Playful barriers-challenges.

- Creating competition within the classroom or with other classes or even with the teacher.
- Ways of comparing and reflecting on personalised performance such as achievements, points, strengths, weaknesses, and ways to reflect on their performance and compare with others.
- Levels, checkpoints, and other methods of progression for "bragging rights".
- Learning badges instead of points or grades.
- Role-play or simulation activities.

Gamification triggers real, powerful human emotions such as happiness, intrigue, and excitement, and it leverages the individual desires for status, achievement, competition and to be part of an inclusive social community to increase engagement [17]. Doing so, it transforms the boredom of executing ordinary everyday tasks into exciting quests. Moreover, it encourages socialisation, peer learning and collaboration or even healthy competition among the users, driving up engagement levels. Therefore, the use of a gamified web-based application can encourage and motivate learners, whether adults or not, towards the change of lifestyle but will also show that learning can be fun and foster continuous personal development [18]. To this purpose, also contributes to the fact that students or users in general will feel that have ownership over their learning, and feel more comfortable in learning this way, which leads them to be more proactive and open to making mistakes and learning from them [15].

Several studies point in the direction of showing games and gamification as very efficient education tools, even posing advantages when compared with other types of digital and online platforms and resources. In their research about The Role of Serious Games, Gamification and Industry 4.0 Tools in the Education 4.0 Paradigm, Almeida and Simões also touch on the point that digital platforms offer interactive ways to engage students in virtual environments, fostering experiential learning that underscores the consequences of individual actions on the environment [19]. Boragine, in a paper exploring the Green Game Frame (GGF), a framework that can be used to identify and select games to educate about sustainable development, advocates for the integration of games and gamified tools in education for sustainability, and even defends that they should be prioritized "[...] among the various educational tools for sustainable development" [20]. This added value lies, according to the author, in the fact that gamification, by providing a hands-on and immersive experience, has been proven to facilitate the understanding of "[...] complex ecological problems and sustainability issues" and expand learners' knowledge and intellectual abilities, by helping them "[...] grasp the interconnectedness of environmental, social, and economic factors in sustainable development" [20]. According to the conclusions of the study, learning through games of gamified experiences also enhances users' awareness of their own impact on the environment and society, fostering, thus, self-responsibility and self-implication in the cause.

Nevertheless, games and gamified digital applications are not only viewed as powerful knowledge and awareness-building opportunities, but are in fact—and most importantly—considered quite relevant in their role of bridging the aforementioned gap between theoretical knowledge and consciousness and action, that is, it is believed that they facilitate the application of knowledge into practice, actually having an impact in changing behaviours and, consequently, improving peoples' lifestyles into more ecological and sustainable ones. Research by Akthar et al. [21] suggests that education significantly influences pro-environmental behaviours, demonstrating that knowledge acquisition through formal education positively correlates with the adoption of sustainable practices. As the global availability of gamified apps and digital tools related to sustainability education has increased, so have the studies aiming at assessing their benefits and the real impact they might or not have in raising, on one hand, awareness and knowledge regarding the topic, and fostering concrete, real-life ecological action. A comprehensive article published by Boncu, Candel and Popa [22] reviewed 29 studies published on this topic to analyse the use—and effectiveness—of serious computer games and gamified mobile apps to foster

pro-environmental information, attitudes, and behaviours. They looked at several types of applications, some of them aimed at facilitating ecological mobility, others at decreasing water and energy consumption, monitoring and changing food consumption to a more environmentally friendly pattern, and so on. While not always straightforward to measure, the benefits of engaging with this type of application seemed to be visible in most interventions, and many of them led to significant and long-term effects among the users. The authors align with others when focusing on the barriers "that prevented the users from previously participating in such behaviours" and stating that these apps and games are likely to "activate the psychological mechanisms that are related to stronger involvement in these actions", thus helping bridge the gap between awareness and action [22].

However, despite the growing recognition of education's transformative potential, the incorporation of sustainability subjects within the conventional curriculum remains limited. This scarcity underscores the urgent need to invest in educational resources that complement existing practices, facilitating comprehensive instruction on environmental issues, ecological stewardship, and sustainable lifestyles. Almeida and Simões highlight that addressing the gaps in the current educational approach demands multifaceted efforts and underlines the importance of interdisciplinary collaboration, advocating for the inclusion of environmental education across various subjects [19]. By seamlessly integrating sustainability concepts into mathematics, literature, and even physical education, students can grasp the interconnectedness of ecological, social, and economic systems.

## 2. Context and Development of the GoBeEco Project

### 2.1. Needs Analysis

The development of the GoBeEco application (1.0) is, undoubtedly, part of a larger effort of applying technology and digitalisation to educational purposes, and, in particular, to education to sustainability. There are many options available in the market, created either by renowned companies or born in the realm of projects and educational initiatives such as this one, but the majority of them seem to focus on tackling a single ecology or environmental issue, instead of adopting a holistic approach to sustainability, where all aspects of human life are considered. This segmentation can lead to an overly narrow intervention, and dismiss the consideration of other topics, and, as they converge on a unique problem, they might require some previous motivation and eagerness of the user to work on that specific topic. Therefore, these kinds of apps can perhaps be less attractive to individuals who want to change their behaviours and become more sustainable but do not know where to start and can feel to specific to be adopted on a daily basis. Common examples of this type of tools include water or energy consumption monitoring and saving applications, carbon footprint calculators, and applications that aim at reducing consumption or making it more sustainable (such as applications for second-hand shopping) or at reducing waste, either at the food level or other. In an approach more similar to GoBeEco's, we can find Zero, which challenges users to reduce their waste and live more sustainably in several aspects of daily life, such as transportation, consumption, food and cooking, and domestic chores. The gamification elements are also similar, as the application includes a point rewarding system for each completed challenge and different levels that the user can climb while solidifying and expanding, continuously, their sustainable behaviour. JouleBug comes also close to the GoBeEco approach—it is a gamified app that encourages users to adopt sustainable habits in their daily lives. It offers challenges and rewards for completing eco-friendly actions such as reducing energy consumption, conserving water, and recycling. Gerber et al. pointed in this direction, stating, after reviewing 115 games falling into the category of "climate" or "environmental", that the majority of them focus on climate change mitigation rather than on adaptation [23]. According to the European Environment Agency, mitigation is essentially related to "preventing or reducing the emission of greenhouse gases (GHG) into the atmosphere to make the impacts of climate change less severe", and, therefore, its strategies are much more related to pollution emissions and their control, "by increasing the share of renewable energies, or

establishing a cleaner mobility system—or by enhancing the storage of these gases—e.g., by increasing the size of forests" [24]. Correspondingly, the greatest amount of games seem to be focused on the topic of energy (36% of studied games), followed by transport (13%) [23], and a smaller portion are games connected to adaptation, that is, with an approach that recognises the already visible effects of the climate crisis but gives preference to taking actions to "prevent or minimise the damage they can cause" [24]. Here, fall games, for example, focusing on behavioural shifts and on actions such as reducing waste and other individual missions. GoBeEco is, then, not only a contribution to the already existing poll of games and tools supporting environmental education, but also an expansion in a direction not so explored, by concentrating its challenges on individual change and action and promoting ecological habits in several categories that go beyond the most common explored in these tools (energy and transportation) and include different priority areas as reflected in the SDGs.

Having this in mind, the consortium wanted to confirm the needs and preferences of the potential uses of the application, and, therefore, conducted a needs analysis with the main objective of understanding the learners' preferences and requirements, that is, what they considered more relevant to learn in terms of environmental issues and how they would prefer to do so. The field research encompassed various dimensions, including an examination of the digital skills and behaviours of educators, the most efficient approaches to addressing environmental concerns, the significance of sustainable actions in energy, water, and mobility, the practices in sustainable energy and water consumption, transportation, and mobility, as well as the prioritisation of sustainable actions concerning consumer goods.

The respondents comprised both adult educators and non-educators in nearly equal proportions. The initial inquiry required participants to designate the environmental issues of utmost importance to them, revealing notable variations in the topics selected. A small majority (55%) deemed as highly significant issues such as the decline or extinction of species and habitats, natural ecosystems (forests, fertile soils), shortage of drinking water, food waste, air pollution, climate change (including natural disasters), and the escalating volume of waste, while three issues—agricultural pollution, food consumption, and greenwashing—were conspicuously regarded as less pertinent.

When asked about the "educational tool that would help them change their behaviours to eco-friendly ones", half of them mentioned "a mobile app with daily challenges on sustainability issues or with daily interesting notifications about sustainability", and a third referred "gamified education contents accessible on a mobile device". Furthermore, an additional recommendation surfaced—the creation of a set of guidelines, such as "10 tips on climate action or sustainable living", as depicted in Figure 1.

These preferences were also reflected in the fact that over 80% of the participants pointed out that they use the internet for professional or personal development, "e.g., by participating in online courses, webinars, or consulting digital training materials and video tutorials". This echoes the priorities that were established by the consortium, who considered that the behavioural change in individual preferences and lifestyles of adults was best facilitated with the assistance of skilful educators who have tools and educational methods most appealing to the learners—in this case, digital and gamified resources and methods—who are attached to their habits shaped throughout the lifetime. In this sense, the needs analysis research confirmed that to increase learners' engagement, educational organisations should offer solutions matching the learners' expectations—82% of learners prefer to learn at their own pace [17] and 70% use mobile applications for this purpose [19].

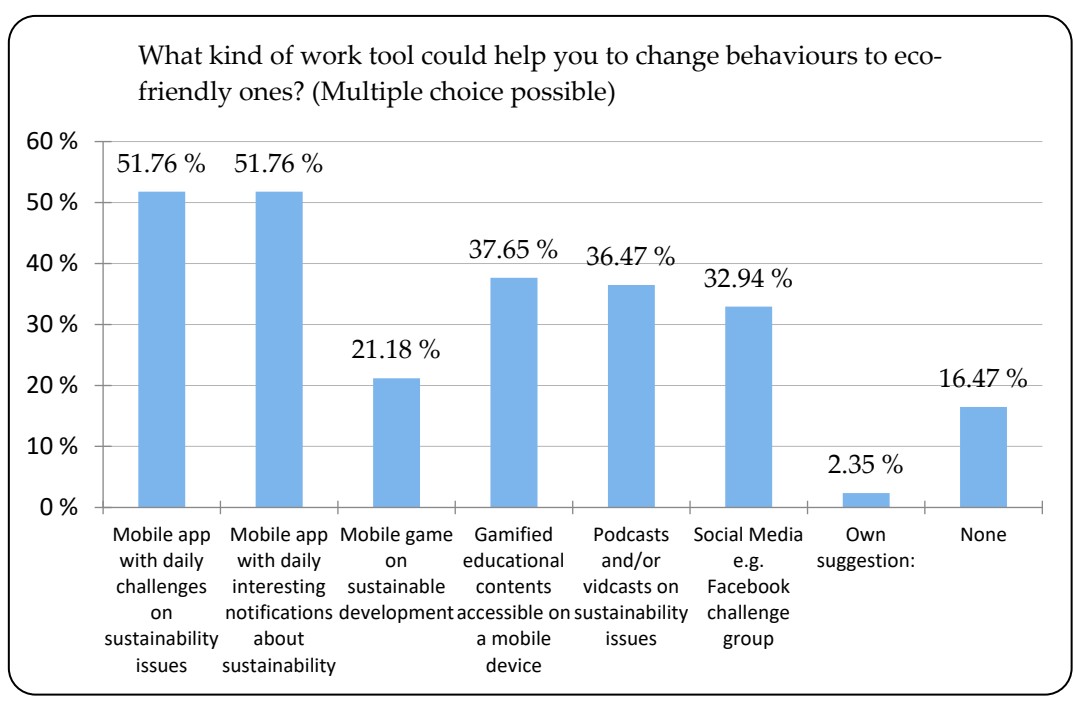

**Figure 1.** GoBeEco Field Research Results.

### 2.2. Methodology Used for the Development of the GoBeEco App

Considering the context, marked by the necessity of educational-related initiatives as an important aspect of citizens' readiness and motivation to act in a more sustainable way, as well as the denoted advantages of deploying digital and gamified elements and tools in those educational initiatives, the GoBeEco project was implemented. The project's main intention was to use the power of gamified applications to foster the forming of pro-eco habits, taking as a starting point the belief that the climate protection and the mitigation of the environmental issues is also an individual responsibility and that it is within each one's capacity to change their ways of thinking and acting in relation to the planet and contribute to a structural change. This focus on the individual and a great commitment to strengthen each citizen's ability to think and act in environmental-related topics led the development of the project, which intended, also, to democratise the knowledge about what can be done, daily, in order to adopt a more ecological way of life.

The partners' decision to develop a gamified app also had in mind the creation of an attractive solution for both autonomous learners and for educators who would want to use the resource in their teaching or training activities. It should also be noted that, by being a digital resource, the app also contributed to fulfilling another goal, that is, the increment of the digital abilities of its users. In parallel, the consortium created a Handbook aimed at capacitating educators to develop and integrate in their lessons a digital curriculum around environmental topics. By doing so, the developed results answered the three central objectives of the project:

- To promote the change of behaviours of adult learners and educators regarding ecology through the change of individual preferences, consumption habits and lifestyles.
- To enhance the development of the digital competencies of educators so that they can inspire learners to understand and adopt those behavioural changes.
- To support educators in developing digital-resources-based curricula and in using it for ecological purposes, by providing them with an innovative step-to-step pedagogical methodology.

Hence, the main result produced in the scope of the project corresponded to the design, development and testing of an application to encourage adult learners to change their lifestyles to eco-friendly ones by gaining practical knowledge to what pro-climate actions

can be done at specific places (home, neighbourhood, shops, office etc) so they take such actions through concrete directions as to what they can do differently.

The development process followed some basic theoretical principles. Firstly, the project proposed the adoption of the concept of Education for Sustainable Development, which describes a holistic and transformative educational process that serves sustainability issues such as climate protection and biodiversity while integrating learning content, learning outcomes, pedagogy, and a digital learning environment in an interactive way to enable inquiry, action-oriented and transformative learning. Secondly, the project's team also privileged the fact that learners of all ages should be enabled to change themselves and the society in which they live. This concept includes "all learning throughout life, which is aimed at improving knowledge, qualifications and competences and within the framework of a personal, civic, social or employment-related perspective. This approach takes into consideration that, especially in times of rapid social, ecological, and economic changes, lifelong learning creates the prerequisites for these, competencies of learners in the sense of shaping society. Thirdly, the project favoured the use of participatory methods to develop critical thinking, collaboration and leadership, teamwork, and other skills. In addition, the international scope of the project intended to enable users to behave as citizens of the world not limited to national borders.

In terms of distribution of responsibilities, Virtual Campus coordinated the overall development of the GoBeEco gamification, monitoring all partners' work and designing the gamification structure, defining mechanics, dynamics, components, aesthetics and storyline (this strategy is presented in detail in the next subsection of this article). PAIZ was the partner responsible for the web-based application development and maintenance work. Regarding the creation of the content and the gamification activities design for all the tasks, challenges and missions of the application (see Section 2.4), as well as all translations and evaluation tasks (project tasks O1/A1, A2, A3, A4, A6, A7, A8, A9), all partners from the consortium participated actively. External participants, belonging to the target group of the project, were involved, in particular, in the need analysis stage (described in Section 2.1), in the different evaluation phases (described in Section 2), as well as in the final events/infodays organised in each country participating in this Erasmus+ project.

*2.3. Gamification Strategy*

The project gamification strategy was based on the Octalysis gamification framework, proposed by Yu-Kai Chou [25]. This framework proposes, as seen in Figure 2, eight significant drives or levers (meaning, accomplishment, ownership, scarcity, avoidance, unpredictably, social influence and empowerment) and relates them with gamification elements that should be included in the gamified learning strategy for maximum impact. This way it is possible to maximize the impact of gamified learning on skills and habits forming as well as in the change of thinking patterns and lifestyles. Adults can not only learn what can be done to help the environment on a global scale but will also get immediate hints and instructions on what they personally can do to contribute, and they will get immediate feedback on their progress. The use of digital gamification tools contributes as well to improving digital skills.

It should be noted that a good gamified system does not need to have all the Core Drives, but it does need to fulfil well the ones it does implement, which means, it must be very well adjusted to the target group drives and motivation. Following the needs analysis, partners debated each core drive and defined, the most relevant ones for the purpose of the project, accomplishment, epic meaning, and social influence. The three core drivers chosen were then unfolded into the gamification elements. For each gamification element, the motivation for the user, as well as a small explanation of how to include the said element in a gamified tool was developed, as seen in the table below (Table 1):

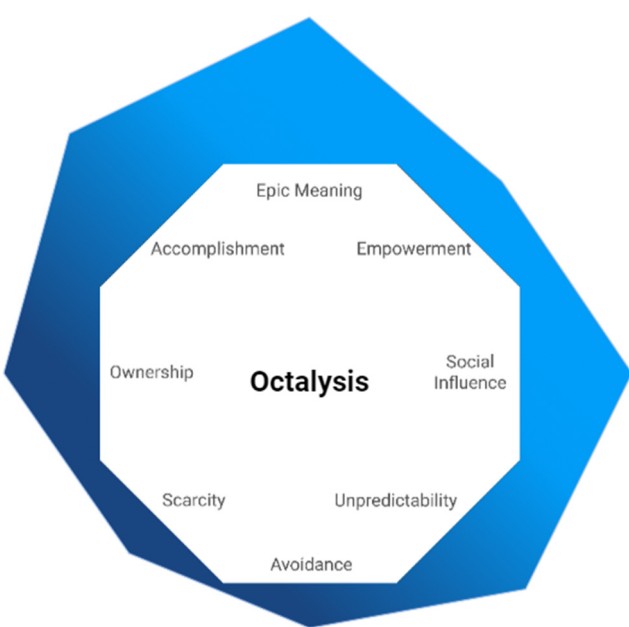

**Figure 2.** The main levers of the GoBeco gamification strategy.

**Table 1.** GoBeEco Gamification Elements.

| Core Drive | Element | Motivation | How to Use |
|---|---|---|---|
| Accomplishment | Quest lists | A quest is a journey or expedition where challenges are overcome. It gives a path for progress for the user | The quests will be organized in missions, challenges, and tasks. Users should have the list of missions, challenges, and tasks available (with an indication of the ones that have been completed) |
| | Status Points | Gives the user the feeling of reward for completing a challenge | Assign status points for completing tasks. Number of points depending on the difficulty of the task (easy, medium, hard). |
| | Progress Bar | Allows the user to situate him/herself in the journey. | Show the progress of the user towards completing a task, challenge, and mission. |
| | High Five | Gives the user the feeling of being recognized. | Greet the user for some additional aspects. For instance, logging in five days in a row, completing n tasks, etc. |
| | Leaderboard | Allows the user to measure his/her own progress in relation to the others. Gives the feeling of public acknowledgement. | Make a leaderboard available allowing to compare the individual performance to other users |
| Epic Meaning | Humanity Hero | Gives the user the feeling of involvement in something that will contribute to improve the entire world. | In the description of tasks, challenges, and missions, reinforce the idea that the user is contributing for the global good. |
| | Narrative | Gives the users a context for the process, namely in terms of the higher meaning. Provides consistency and connection to the different missions. | Provide a narrative to contextualize the missions |

**Table 1.** *Cont.*

| Core Drive | Element | Motivation | How to Use |
|---|---|---|---|
| | Friending | Allow users to relate closely with other persons. | Allow users to relate closely with other persons. |
| | Gifting | Allow users to interact with other users fostering his/her generosity. | Allow users to interact with other users fostering his/her generosity. |
| Social influence | Social prod | Gives the user the feeling of social recognition. | Allow the user to show on social media his/her achievements |
| | Group quests | Allow users to relate and interact with others, work, and feel part of a team. | Allow users to collaborate in completing a task. |

### 2.4. The GoBeEco App: Final Version

The GoBeEco application was designed taking into account the aforementioned objectives and considerations and embarking on the chosen gamified strategy and elements, with the aim of functioning as a daily self-standing resource that users could resort to adopt sustainable habits in several areas of their lives and track their evolution in that same process. First, and considering the belief that an ecological transition must encompass all the most relevant areas of a person's life, it was decided that the app would cover several dimensions of one's life, each one of them corresponding, in the tool, to a separate "Mission". This led the consortium to divide the activities into five missions (see Figure 3): (1) "Public space—individual responsibility", with several tasks aimed at promoting ecological habits in collective, shared spaces—such as parks, streets, and others-, (2) "Your home—your decision", dedicated to promoting change at home, concerning actions such as cooking and cleaning; (3) "Getting and using consumer goods", with activities challenging consumers to make conscious and better-informed buying choices; (4) "Behaviour at work", stimulating workers to bring change to their offices or places of work and (5) "On the green way", a mission with the objective of promoting the adoption of greener mobility choices. The app includes a "Special Mission", only accessible after the completion of all others, versing activities related to recycling, repairing and reusing materials, and the promoting of "do-it-yourself" (DIY) projects.

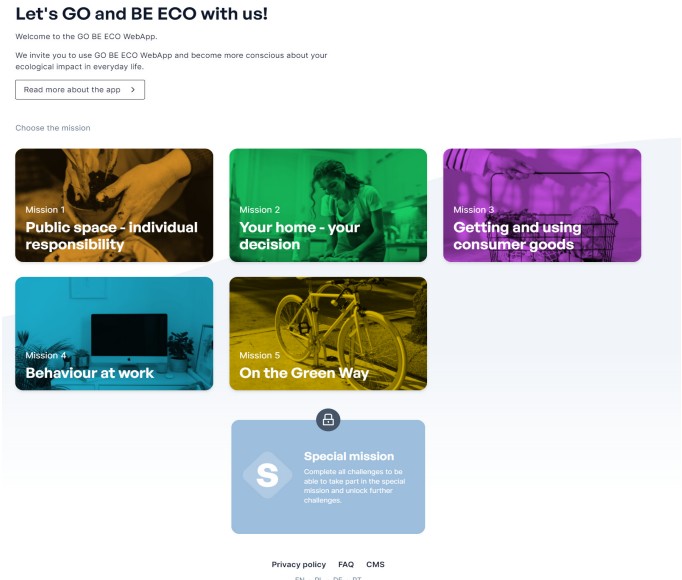

**Figure 3.** Screenshot of the GoBeEco initial page, with all the missions.

Each mission was divided into four or five challenges, which can be defined as a set of activities related to a wider theme within the mission. Each challenge was then further divided into a variable number of tasks (see example in Figure 4). These tasks are the small activities that users are supposed to be gradually completing, on a daily basis, until they become integrated part of the routine. In total, the app includes over 100 tasks, which are designed to take up until one month to complete in their totality—it is believed that, after this time, the user will already be quite familiar with what one can do autonomously, in several areas of live, to increase our sustainability level. The tasks were designed with different levels of difficulty and the user will score different points for their completion, depending on the level. The completion of the task is signalled by the users themselves—there is no other way of tracking their conclusion-, as this tool was thought as a way of promoting self-reflection, awareness, and autonomous action, which is not compatible with a more strict, formal way of assessing the completion of the tasks, challenges and missions. However, as to promote further assimilation of knowledge, at the end of each mission, users face a short quiz, whose successful completion is rewarded with a badge. With the purpose of encouraging and motivation users to continue their efforts, the application also has a ranking system integrated, which allows people to assess their position on a global scale of competitors.

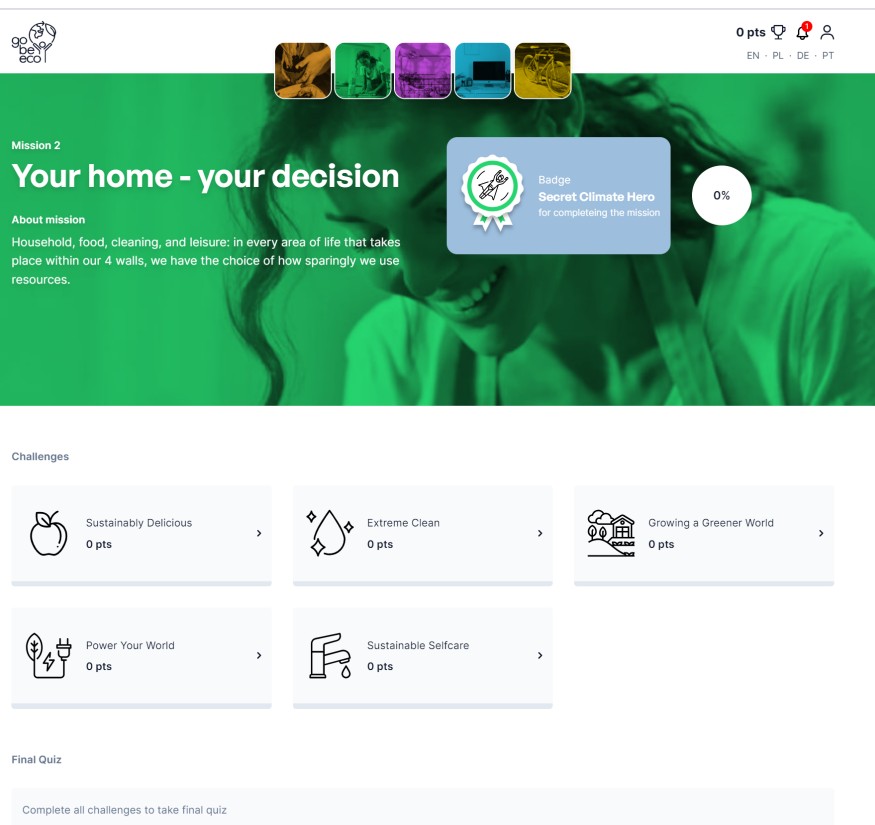

**Figure 4.** Example of the challenges contained within the "Your Home—Your Decision" mission.

## 3. Materials and Methods

The materials and methods used for the analysis of the impact of the GoBeEco application on promoting sustainable habits and pro-ecological behaviour were aligned with the impact and success indicators set by the Quality Assurance responsible partner, alongside with the coordinator, right at the beginning of the project. In specific, the validation and evaluation of the GoBeEco application comprised three main phases, as in can be read in Table 2, which details the parameters set, the number and profile of participants to be reached in each validation and evaluation phase, and the method to the implemented:

Table 2. Validation and Evaluation Criteria and Methods for the assessment of the GoBeEco application.

| Evaluation Phase | Quality Criteria | Quality Indicators | Evaluation Methodology/Data Sources |
|---|---|---|---|
| **Activity 6 and 7 (Phase 1)** | Gamification in EN language tested and positively assessed in terms of congruity of contents & activities | min. 25 learners from partner countries confirmed that all mechanisms are in place and Gamification serves the objective | Feedback gathered through questionnaire |
| | Successful user experience & usability testing of the Gamification | At least 25 learners confirmed that:<br><br>- Selected mechanics & processes are functional<br>- Gamification is compatible with the computing & mobile<br>- Environment (computing capacity, mobile & PC browser compatibility, etc.)<br>- Gamification is user friendly | |
| **Activity 9 (Phase 2)** | Gamification in local language positively assessed by sufficient number of pilot educators outside the partnership | At least, 50 learners + Partners staff tested the gamification in PL, DE, PT; 75% of learners declare that gamification impacted behavioral changes regarding ecology, increased their pro-ecological awareness, is engaging & motivating | Feedback gathered through questionnaire (the same as in the previous phase) |
| **Activity 10 (Phase 3)** | Quality of the Gamification in local languages is confirmed | 13 partners' educators confirm that Gamification fits the required quality standards and is ready to use by a wider audience | Feedback gathered through the questionnaire and in the focus group |

It should be noted that, according to the quality indicators set, a minimum of 25, 50, and 13 users, respectively, should have been involved in each testing phase; however, this refers to the total numbers to be reached by the partnership as a whole, which means that each partner (there were 6 in total), was responsible for gathering, at least, the feedback of 5 users in the first phase, 10 in the second, and between 2 and 3 in the third step of evaluation. Therefore, the numbers presented here are in accordance with the parameters developed by the project's coordinator and quality assurance leader and pertain to the pilot testing evaluation conducted in Portugal by the assigned partner, Virtual Campus. Regarding the profile of the participants involved in the piloting stages, this coincided with the target groups established as early as in the project's application:

- Educators of adult learners from partner organisations;
- Adult learners willing to get knowledge and change their habits regarding pro-eco behaviours;
- Partners organisations' staff;
- Relevant stakeholders outside partner's organisations at local/regional/national/EU level such as National and European adult learning and environmental agencies, organisations and authorities responsible for systems and policies concerning environmental protection.

Each partner was responsible for defining the most adequate media to reach the potential participants, either by using their own websites and social media channels, resorting to online adult education groups and forums to gamification and e-learning forums, engaging in e-mail campaigns using their own databases and networks, reaching to lifelong education and e-learning conferences, educational fairs or gathering interested parties through dissemination actions. Given that all partners counted with extensive networks

from previous projects and many of them had established contacts with organisations relevant to the project (and, also, some of them are connected to educational activities, which facilitates the access to the target groups), they had no major difficulty in involving participants, who were selected according to their relevance to the project and likelihood of influencing other participants (e.g., as role models, through education/training or at the policy level).

The first phase was conducted with English-speaking participants (as the tool had not been translated yet at that point) and comprised, itself, two steps. Firstly, partners should contact the participants and, if needed, organise a workshop presenting the GoBeEco project and app, as well as providing them with all the contents and tools required to perform the evaluation. Then, participants should be given some time (three weeks) to test and use the app on their own; being this time up, they were asked to evaluate the application they have explored through answering a previously prepared questionnaire (Appendix A.1). This same questionnaire was later applied in the second phase of evaluation, where users were now expected to test the national (Portuguese) version of the application and check, apart from technical issues and adding to the reflection on how the app had contributed to changing their ecological behaviour, pay close attention to any language-specific issues. Through the distribution of this questionnaire, all together in phase one and two, seventeen (17) respondents were involved. For this purpose of this article, we will not consider the answers to all questions, but only those which are more relevant to its aim; that is, the answers to the questions that are related with pro-ecological behaviour and pro-ecological change. As it can be consulted in more detail in the appendixes to this article (Appendix A.1), participants were asked to rate each several statements, including the ones that follow, from 1 (Strongly Disagree) to 5 (Strongly Agree):

- I have positively changed my behaviour concerning sustainability by using the app.
- By using the app, my awareness of ecological contexts has improved.
- By using the app, I am motivated to engage more intensively with the topic of sustainability in my life.
- I have been able to expand my knowledge of sustainability in the long term.

Later, the final evaluation phase (phase 3) was organised, consisting of a focus group and a final survey evaluation, more focused in assessing the gamification used, more specifically, if the gamification was ready and up to the declared quality, serving the purpose it is meant to serve. This evaluation was also meant to verify the impact the gamification has on promoting and inciting pro-ecological behaviour. The questionnaire used for this validation stage can also be consulted in the Appendix A.2. For the purposes of this article, we will focus on the impact of the gamification elements in the adoption of a more conscious and ecological behaviour. That said, the analysis will be centred on the Section 2 of the survey, "What experiences did you have with the GoBeEco Gamification?" (Appendix A.2), concretely, on Yes/No questions that follow:

- The use of GoBeEco Gamification positively influenced my ecological behaviour.
- The GoBeEco Gamification raised my awareness for ecological contexts.
- By using the GoBeEco Gamification I am motivated to continue working on a more environmentally conscious behaviour.
- Do you have fun using the Gamification?
- In GoBeEco Gamification, appropriate areas (Missions/Challenges) help users improve their daily actions.
- In GoBeEco Gamification, clear habits are named, and the app offers good guidance on how these habits can be made ecologically better.
- GoBeEco Gamification provides practical knowledge about what can be done for the climate in specific places I visit.
- GoBeEco Gamification encourages different types and generation of users to take long term action and develop an understanding of the importance of developing environmentally friendly habits.

- Do you feel now you could act as a change agent, to encourage others to develop themselves?
- Do you think it is worthwhile to go back to the gamification?
- Will you recommend the Gamification to others?
- Do you think, after using the Gamification, you indirectly inspire others to change habits?

Additionally, in this stage, a focus group was organized. To guide the discussion, the participants taking part in this phase of the evaluation were provided with, before the online gathering of the group, with an Empathy Map, shared in a collaborative platform (Jamboard). They were asked to leave their feedback regarding the GoBeEco application according to the four quadrants which compose an Empathy Map, and that intend to lead the user to answer to what he/she did, said, thought and felt regarding the use of a particular product:

- Say—What the user says about the product. Ideally, this section contains real quotes from users recorded during interviews or user testing sessions.
- Think—What is the user thinking about when interacting with a product? What occupies the user's thoughts? What matters to the user?
- Feel—This section contains information about the user's emotional state. What worries the user? What does the user get excited about? How does the user feel about the experience?
- Do—What actions does the user take? What actions and behaviors did you notice?

In this project, partners followed the structure of the map as originally created by Dave Gray, the founder, and chairman of Xplane—with the objective of having a tool to implement in the design process, after the initial research but before the ideation stage—and as widely used by designers and creators and adapted it to the purposes of the GoBeEco project (see Figure 5).

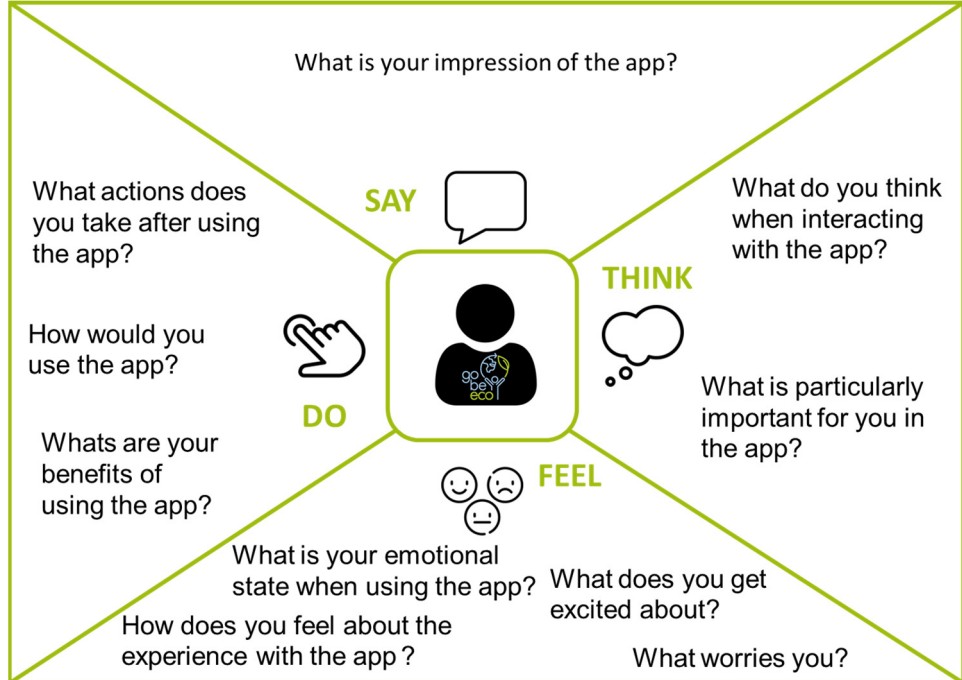

**Figure 5.** Empathy Map used in the focus group.

In total, the evaluation in this country involved twenty-four (24) participants, among who seventeen (17) participated in phase 1 and 2 altogether answering the evaluation questionnaire, four (4) participated in the third round of evaluation and another four (4) took part in the focus group, also organised during this phase. Therefore, the next

section will dwell on the answers shown in Table 3 which refers to the participation of the seventeen mentioned participants, on the answers presented in Table 4—which displays the four participants in the questionnaire shared in phase 3, and on the qualitative responses gathered during the focus group.

**Table 3.** Distribution of answers by position on the Likert Scale (nominal numbers).

| Statements Regarding the Improvement of Ecological Awareness and Change of Behaviour | Strongly Disagree (1) | Disagree (2) | Neutral (3) | Agree (4) | Strongly Agree (5) | % Agreeing/Strongly Agreeing |
|---|---|---|---|---|---|---|
| "I have positively changed my behaviour in relation to sustainability by using the app". | 0 | 1 | 3 | 13 | 0 | 76% |
| "By using the app, my awareness of ecological contexts has improved". | 1 | 0 | 0 | 13 | 3 | 94% |
| "By using the app, I am motivated to engage more intensively with the topic of sustainability in my life". | 0 | 1 | 1 | 12 | 3 | 88% |
| "I have been able to expand my knowledge of sustainability in the long term". | 0 | 2 | 0 | 12 | 3 | 88% |

**Table 4.** Distribution of answers (Yes/No) to the statements presented (nominal numbers).

| Statements Regarding the GoBeEco Gamification Elements and Their Impact on Ecological Behaviour | Yes | No |
|---|---|---|
| The use of GoBeEco Gamification positively influenced my ecological behaviour. | 4 | 0 |
| The GoBeEco Gamification raised my awareness for ecological contexts. | 4 | 0 |
| By using the GoBeEco Gamification I am motivated to continue working on a more environmentally conscious behaviour. | 4 | 0 |
| Do you have fun using the Gamification? | 4 | 0 |
| In GoBeEco Gamification, appropriate areas (Missions/Challenges) help users improve their daily actions. | 4 | 0 |
| In GoBeEco Gamification, clear habits are named, and the app offers good guidance on how these habits can be made ecologically better. | 4 | 0 |
| GoBeEco Gamification provides practical knowledge about what can be done for the climate in specific places I visit. | 4 | 0 |
| GoBeEco Gamification encourages different types and generation of users to take long term action and develop an understanding of the importance of developing environmentally friendly habits. | 4 | 0 |
| Do you feel now you could act as a change agent, to encourage others to develop themselves? | 4 | 0 |
| Do you think it is worthwhile to go back to the gamification? | 4 | 0 |
| Will you recommend the Gamification to others? | 4 | 0 |
| Do you think, after using the Gamification, you indirectly inspire others to change habits? | 4 | 0 |

## 4. Results and Discussion

As aforementioned, as this article's objective is to understand the positive impact that applications can have in promoting ecological behaviours and in increasing awareness regarding this topic, the focus in this section will be on the questions from the questionnaires (Appendix A.1) which versed these aspects. Therefore, other answers regarding, for example, the technical and design assessment of the application will be disregarded here, as they do not align with the major aim of the current publication.

Table 3 compiles, as mentioned before, the answers to the same questionnaire (distributed both in Phase 1—project activities A6 and A7—and in Phase 2—project activity A9), aimed at understanding, after a period of autonomous exploration of the application by the users, the impact it had on both their awareness towards sustainability and their actual changes to more ecological habits and practices. The results are presented below:

The table compiles the answers of the 17 participants in this survey for each one of the sentences presented. In each one of the statements, we can see that the greatest concentration of answers lies in the "Agree" option: thirteen of the seventeen respondents

chose that option in the first two sentences presented here, and twelve out of seventeen in the latter two. When making a sum of all "Strongly Agree" and "Agree" answers received for each statement, we can see that the overall evaluation was overwhelmingly positive, as, for each one of them, the percentage of people stating their accordance to the positive claims was pronouncedly over 50% and, in some cases, even close to 100%.

As it can be observed, all statements received a similar rating, with the majority of participants agreeing, or strongly agreeing, that the app somehow contributed to a change in their behaviour, that they became more ecologically aware, and they were able to expand not only their knowledge but also their motivation to be more sustainable in the future. The topics that seemed to be less consensual were "I have positively changed my behaviour in relation to sustainability by using the app" and "By using the app, my awareness of ecological contexts has improved", which can be perhaps explained by the fact that most people didn't use the app long enough to register major changes in their behaviour. Another possible explanation can be that the app is focused on action and doesn't provide extensive theoretical information, which could have let some participants rate the second statement slightly more negatively. Nevertheless, even considering that the application was tested for a limited period of time, it can be observed that it was able to fulfil its purpose and already incite behavioural change and improvement.

The questionnaire results highlighted, thus, two main effects of the project's results on the involved participants: on one hand, positive behavioural change, which was recorded by the great majority of respondents and, one the other, motivation and expansion of knowledge, two consequences reported by the users of the webapp, which illustrate, mainly, the potential that the tool has to disseminate environmental knowledge and sparkle long-term motivation to engage in sustainable practices, which, in turn, can be an important success and impact predictor for the future of this outcome, as it can not only provoke short-term change but also to capacitate users with the knowledge, competences and inspiration necessary to continue pursuing that line of behaviour and self-actualise their level of compromise and expertise. This also evokes the application's twofold potential. On one hand, the fact that the majority of the participants in this evaluation phase stated that the tool had a positive impact in raising awareness and knowledge (second and fourth statements), which was aligned with what was discussed in Section 1.1, regarding the educational potential of digital environments and game-based strategies. On the other, as a significant number of participants considered that the tool developed in this specific project had the power of "changing behaviour" (first statement) and motivating them to "engage more intensively" (third statement), we can see how the application isn't limited to building that knowledge, but also stimulates its practical application. In this sense, the GoBeEco application falls into the "adaptation" category of environmental games, as defined by Gerber (see Section 2.1), because it focus on behavioural shifts and on actions such as reducing waste and other individual missions; however, somehow, and more indirectly, it also expands the concept, because it offers more than guidelines to a certain behaviour or action—it also capacitates users to understand which are the major environmental problems affecting our lives right now, showing the importance of both "mitigation" and "adaptation" strategies.

For the sake of the article scope, it is also relevant to consider the participants' qualitative comments, left in the open section of the survey. Based on their answers (and here considering, as the most crucial for our purposes, the ones regarding the impact of the application on their real-life attitudes and habits), we can distinguish different categories of answer, each one of them stressing one benefit/impact related with the regular use of the application. The following findings provide examples taken verbatim from participant surveys, while the theme titles represent our interpretation of the data. It should also be noted that each piece of feedback is identified by the number attributed by the user mentioned it (Fx), as well as the phase to which the feedback belongs (Phase 1 or Phase 2, as explained before).

### 4.1. Increased Knowledge, Reflection and Self-Awareness

One of the primary effects that the application seems to have in users, and which is aligned with some of the central benefits of this kind of approach, as discussed before, is an increase in knowledge about sustainability and about the effects of one's actions have on the environment—both from a positive and negative point of view: "I was more conscious about my daily life and how much it impacts the environment" (F1, Phase 1); "It made me reflect on my daily habits"; "It made me think about the consequences of my everyday behaviours" (F2, Phase 1). This rise in awareness and knowledge about our individual role might also suggest that even if and when people have knowledge about the environment and ecofriendly actions, they might still lack some specific knowledge about the role that individual changes might play and often dismiss their impact potential, which can, perhaps, be considered as one factor hindering the closing of the gap between awareness and action, as people get stuck on the lack of knowledge on how to proceed, and, even if they know, they might think that it doesn't make a real difference: "The app showed me how easy it is to live more sustainably. I didn't know that before" (F1, Phase 2); "I would not have thought that it was so easy to live ecologically" (F2, Phase 2).

### 4.2. Increased Motivation to Be More Sustainable

Another recognized advantage of the use of games, applications, or gamified tools is the increment of motivation and willingness to change and behave more sustainable. It could be said that motivation is one important factor—even if not sufficient by itself—that stands in the middle of awareness and action, and, when efficiently stimulated, can have fruitful results. The participants in the survey also highlighted this aspect was one of the most valuable of the app, stating "The main benefit of using this app is that my awareness of ecological issues and problems increased and it gave me the motivation to change something in my life, even if it is just a small thing" (F3, Phase 1) and "Using the web app strengthens and motivates oneself to behave in a sustainable manner" (F3, Phase 2).

### 4.3. Empowerment, Sense of Control and Sense of Pride

As mentioned earlier, sometimes knowing what to do, being awareness of the potential effects of our actions, and being stimulated to do so by some kind of encouragement is not always enough, which leads some authors to defend that sustainability applications and tools should focus on the "psychological mechanisms that are related to stronger involvement in these actions" [22]. This is very helpful, as we should not discard the way people feel when doing something, because taking it into consideration and guaranteeing that they feel good is a great way of ensuring that they will keep on doing it. Some participants reported feeling empowered: "It made me feel that I can help the planet with my attitudes" (F4, Phase 1) and with a sense of increased pride on themselves: "It made me proud that I am ecological" (F5, Phase 1). This is connected also with another crucial factor that is known as contributing to people changing behaviours and habits—the social effect. It has been proved that the great majority of people are influenced by the group and community effect—that is, that they are likely to do what other people are doing: "Telling online shoppers that other people were buying eco-friendly products led to a 65% increase in making at least one sustainable purchase. Telling buffet diners that the norm was to not take too much at once (and that it was OK to return for seconds) decreased food waste by 20.5 [26]". The justification is not as simple as "we want to copy others" but lies in the recognition that we are societal beings eager to be part of groups and of communities, and to feel that we belong to something bigger than us—either that be Earth salvation or a community of other type. This is recognisable in some participants' comments, such as "My eco-awareness and eco-habits increased, as I am much more sensitive to what others are doing as well" (F6, Phase 1).

*4.4. Change of Behaviour and Call to Action*

Finally, and as the ultimate advantage of this apps, and, in particular, of GoBeEco, is the call to action and impact in changing behaviour, as this represents the main objective of the project. It should be reminded that the focus of the initiative was to capacitate users to day-by-day, through a cumulative and constructive perspective, to change their lifestyle, but always giving preference to simple and quotidian actions, in order to instil the notion that sustainability can be adopted through a gradual approach and that there are accessible and democratic ways of being part of it. In this sense, we can state that the application reached it purposes: participants referred to its usefulness in making them act in a more eco-friendly way: "I would use app at least twice a week just to remember about ecological ways of doing daily things and I would hope this would become one of my good habits" (F7, Phase 1).; "I am doing a lot more of eco-things described" (F7, Phase 1), but also mentioned specific actions that the application inspired them to do: "It made me take some concrete actions: buying a soda streamer; leaving water for the birds, abstaining from picking flowers for fresh bouquets; I was reminded to go back to carrying my own water bottle; I made DIY accessories for the garden out of old stuff; I am looking for bamboo towels; I remember to turn off the light, I boil only enough water in the kettle; I take my garbage with me when I am in the public space; I delete old e-mails" (F8, Phase 1).

That said, it can be stated that this period of testing allowed the consortium partners to gain a deeper insight into the GoBeEco app and provided them with some detailed feedback that truly helped them understand the real impact and relevance of the product developed, which was also very useful to implement useful and important changes. In general, the feedback was incredibly positive, with most of the participants claiming that the app was easy to use and with interesting elements, that it was visually appealing and that the themes approached were urgent and interesting. Despite some room for improvement, and on a positive note, all participants seemed very dedicated and interested in the app and believed that it had great potential to be used in the future as an engaging and innovative educational tool, capable of changing people's habits and perceptions about the environment.

The third phase of evaluation, as explained in the previous section, was a crucial one, not only because it allowed the consortium to collect further feedback about the application, but also it gathered some selected educators who had been involved before in other phases—and were, thus, familiar with the tool and. They were also especially important agents to allow partners to evaluate some aspects through users who had been in the process of exploring the GoBeEco project, which favours an analysis focused on continuous development and learning. This was relevant because, in this stage, partners were committed to assess, in particular, the impact that the gamification elements included in the application might have on fostering ecological awareness and behaviour, in particular in users that had already used the tool developed and had stated that it had had impact on their lifestyles—the idea was then to understand what role, if any, the gamification played in that impact. To this purpose, the following questions, from the complete survey set (Appendix A.2), and its respective answers, will be considered:

While recognising that the sample of users answering this questionnaire was relatively small, the idea was, as referred, to specifically focus on the role that gamification played in the pro-ecological and sustainable change registered in beneficiaries that were already using the application and had stated a positive influence and concrete changes in their lives. As it can be seen in the table, all the four educators participating in this evaluation stage in Portugal considered that the gamification elements were an important part of the webapp created, evaluating positively the overall effectiveness and attractiveness of the tool. Users reported that the gamification positively influenced their ecological behaviour, raised their awareness of environmental contexts, and motivated them to continue environmentally conscious actions. They also expressed enjoyment and engagement with the gamification elements, highlighting the effectiveness of gamified experiences in sustaining user interest and in encouraging long-term action among users from diverse demographics to act as

change agents, inspiring others to develop environmentally friendly habits. The distribution of this survey, more focused on the gamification elements of the webapp, helped reinforce, also, the consideration that digital tools—in this particular case, applications—can in fact have a positive impact in fostering environmental awareness and, more, to bridge the gap between that awareness and real ecological change and actions, but also that when gamification is integrated in those same tools, the satisfaction and engagements levels of users seem to improve, as well as the positive outcomes of using the application. The impact is, then, connected with both the existence of these tools and the way they incorporate game-based elements; these two dimensions seem to be well connected.

The feedback gathered during the focus group sessions reinforced the data previously collected through the surveys, but also expanded it, by disclosing additional positive outcomes of the usage of the webapp, namely: increased consciousness and reflection, as participants emphasized heightened awareness about daily actions impacting the environment; a sense of pride and achievement, as the app propelled them to consider their daily actions and consequences and led them to adopt better habits which consequently was translated into an increased sense of pride and achievement; and, lastly, the intention to continue using the tool, as many participants expressed their intent to continue using the app regularly, integrating it into their routines as a reminder of ecological practices.

The findings of the several phases of the pilot evaluation of GoBeEco are aligned with similar studies that had succeeded in attesting the positive role and impact of digital applications and gamified tools in promoting sustainable behaviour. Some investigations, such as those conducted by Guillen-Hanson et al., highlight general benefits in engaging with this kind of tools and platforms, stressing the overall advantages they carry in motivating and facilitating people's transition into more environmentally friendly lifestyles, particularly when these applications integrate some gamification elements and features. This is aligned with the conclusions drawn from the first two evaluation stages of GoBeEco (Tables 1 and 2) combined with the feedback gathered through the third stage (Table 3); that is, corroborates that while these applications seem to have indisputable environmental benefits, these are ensured, and perhaps enlarged, by gamified elements included [27]. More, another study conducted by Miao, Saleh, and Zolkepli shows a mutual influence between what happens within the application and the real world, in a two-fold way. As the authors show, it all starts with the Alipay Ant Project, a "game" integrated into Alipay, a Chinese renowned online and mobile payment platform, widely used for the purchase of numerous goods and services. This game rewards its users with "green energy points" each time they take a step to reduce their emissions, such as by biking to work, going paperless and buying sustainable products; then, these green energy points are transformed, at some point, into a virtual tree on the user's app, which the company matches by planting a real tree or by working towards the conservation or protection of some endangered area. Here, the process of promoting environmental behaviour through digital means starts by inspiring users, through a digital tool, to carry sustainable actions in the offline world. This, in turn, has an effect on the app—the growth of a virtual tree, which functions as a motivating element—but it subsequently impacts the real sphere once again, as the company is compromised in planting actual trees. This is already sufficient to attest the circular process of how digital tools, particularly those anchored in gamification elements, inspire users to act in positive way towards the environment, but the relation becomes even stronger, as authors of the study found that, after the company has planted the trees, this reverts once again to the users, as they are, then, more likely to "[...] continue practicing sustainable behaviours and planting trees" [28]. The continuous interaction between online and real worlds, then, seem to be very fruitful, and could, once again, validate the potential of the GoBeEco application—despite it not offering the same level of online interaction and simulation, it does encourage some exchange, as users are encouraged to perform sustainable habits offline, in their daily lives, and then register them on the app, receiving, from that, some points. The study conducted on Alipay Ant project illustrates how that could be even further explored, by, for example, including some more visual and interactive

representations, in the application, of the good made, or by facilitating a more concrete and visible change in the world through the navigation in the tool.

Other studies state the relevance of these applications in, particularly—and more than inspiring direct change or enhancing a complete change in their users-, contributing to the slow, continuous, and overtime building and solidification of sustainable habits. This is important because it is directly aligned with one of the main purposes of the GoBeEco project and application: to instil personal awareness, responsibility, and compromise, in order to ensure an autonomous, continuous and durable implementation of these habits into the users' lives. Research conducted in the realm of another European project, UrbanWaste, intended to motivate tourists to recycle while travelling, which the objective of addressing a major problem nowadays—the waste management and jeopardised sustainability of highly touristic places, which struggle with an enormous flux of visitors and correspondent waste. However, the intent of the project was also to see how these tourists would behave once they went back home, and how the use of the gamified application during the holidays would impact their long-term behaviours. They came to realise that it could, in fact, have a positive impact; the study suggests that gamification, more than promoting consistent recycling behaviour in tourist destinations, contributed to generating that habit in the long term (after visitors returned home), particularly by fostering both intrinsic motivation (altruistic feeling) and internalized extrinsic motivation (behaviour visibility and destination branding improvement) [29]. Along the same line, another study conducted by Kronisch examined how a gamified behaviour-change app with gamification elements, a standard app, and a webpage influenced participants' sustainable behaviours. Over the course of one month, participants interacted with the interventions and documented their experiences with sustainability and the technology in diaries. What was revealed by the research was that gamification promotes extended engagement, resulting in a beneficial impact on behaviour. The author concluded that it has the potential of enhancing both knowledge about and the inclination to commit effort to sustainable behaviours and continuing doing that in the future [30].

Lastly, it should be noted that we also highlighted the role of the GoBeEco application in promoting the growth of knowledge, as well as the adoption of ecological and sustainable behaviours in several dimensions of one's life, instead of focusing in just one. This is valuable also for the flexibility it favours: on one hand, for holistic approach it promotes, and on the other, because it allows users to focus on the desirable dimension, given that several studies have been proving the multiple advantages these sustainability applications can have in addressing separate problems and fostering better behaviours in several distinct areas of life and of environment protection. This has been shown by Venessa and Aripradono, who illustrated how gamified applications can encourage households to reduce food waste [31], by Doğan-Südaş et al., who inferred that gamified purchasing apps that influence customers' behaviours and lead them to make eco-friendlier buying choices [32], by Pivec and Lu Hsu, who showed the relevance of gamification in favouring ecological practices, in particular, in particular waste separation, energy saving, and water consumption regulation [33], and by Markopoulos et al., who showed how gamification can not only promote pro-environmental behaviours but also improve, and taking sustainability as a broader concept, organisational culture and environment, aligning it with democratic values [34].

Taking this into consideration, and also looking at the reactions, feelings and actions sparked by the usage of the GoBeEco app, it is possible to state that the three main dimensions proposed to be addressed by the gamification elements—accomplishment, epic meaning, and social influence—were successfully tackled, through the completion of tasks, challenges and missions, the acquisition of points and badges, through the sense of belonging to a wider objective of saving the planet Earth (which was expressed in the comments by users mentioning their satisfaction in contributing to a positive general change), and through the engagement with others, both in the context of the activities, which frequently call for cooperation and collaboration with friends, families and colleagues, but also

through the competition within the app. It is also worth noticing that several participants denoted habit changes both in their personal and collective spaces, which is promoted by the missions targeting both individual and communal spheres. This is extremely positive, as it fosters a holistic and multidimensional approach to sustainability, and contrasts what has been debated by some authors as a potential shortcoming of ecological applications: the fact they tend to focus on promoting private-sphere actions, as they follow a "particular model of environmental education", which "[...] focuses principally on the private-sphere environmentalism rather than preparing students for public actions" [35]. It is believed that GoBeEco app and project constitute a positive step into this direction, as they promote, as mentioned before, self-reflection and critical thinking, and, at the same time, provide users with relevant sources to understand the complexity of the problems tackled. In this sense, for all that was mentioned, it is crucial not to underestimate the role of these initiatives, as they allow for the development of skills and, simultaneously, of environmental awareness, which is, in turn "a prerequisite for environmental citizenship behaviour" and favour what is recognized as central to enhance it—the focus on direct, practical and concrete experience [36].

*4.5. Limitations of the Study and Future Research*

Although the present study provides valuable new insight about the role of digital tools and gamified applications in promoting sustainability awareness and behaviour and, specifically, about how this contributed to so in the Portuguese context, something that wasn't very much explored before, it is not without limitations. First, the sample size is admittedly small, which might compromise the wide application of the study and the draw of more solid conclusions. However, it should be noted that the numbers reached are in conformity with the quality indicators numbers defined at the beginning of the project by the quality assurance leader and the coordinator (explained in more detail in Section 3), which were set as equal for all partners in the consortium. Given that this validation was conducted within an Erasmus+ project, where evaluation stages have a limited time and scope, we believe that we have complied with what was the objective of the project and subsequent study. As this article derives from a piloting evaluation stage, the purpose was not to generalize, at least at this point, the findings to a larger population, but rather to test the usability, usefulness and positive impact of the GoBeEco application in users' lives, and understand if it could be a valid and valuable tool to promote sustainable behaviours and the cumulative, daily build of habits. Naturally, this was made in order to infer some general conclusions of how similar applications and tools can act as powerful instruments to fulfil the mentioned purposes, but always keeping in mind that we depart from a specific example and, consequently, sample size—the target groups involved in the GoBeEco's testing phases. Also, the piloting evaluation aimed to explore qualitative impressions and subjective experiences of participants in a context that has not been extensively researched previously, particularly within the Portuguese setting. Given the exploratory nature of our research and the novelty of the topic in our geographic area, our focus was not on obtaining a large sample size but rather on delving deeply into the perspectives of a select group of participants. We tried to overcome these limitations by focusing on the qualitative information received, and build on the participants' comments to understand how they felt about the application and how they considered it to have affected their lives, and, also, by conducting several evaluation phases, which, while not translating into larger samples, functioned as an iterative and cumulative process, allowing us to confirm (or not) the conclusions taken from the previous stage, as we tried to maintain some uniformity regarding the audience involved, to verify how continuous and medium to long-term use could have positive effects. So, it could be said, continuity and repetitive testing was privileged instead of rotativity or enlargement of the sample sizes. While we acknowledge that a larger sample size could potentially enhance the robustness of our findings, we believe that the richness and depth of insights provided by our small but diverse sample offer valuable contributions to the literature. In the future, larger evaluation sessions would

of course be of extreme value, eventually also by expanding the scope of the testing and including European conclusions (involving, then, all the countries involved in the project).

Another limitation of the study lies in the use of self-reporting tools in all phases of evaluation (surveys and guidelines/empathy map for the focus group). As participants were asked to state their own opinions and were the ones responsible for claiming the efficiency, or not, of the application, there is no absolute guarantee that they have not exaggerated or lied, for various reasons, which can contribute to some bias of the study (particularly, considering the social desirability bias). We tried to overcome this issue by ensuring clear instructions on how to interact with the application and about what was expected from participants (one of the steps of the evaluation phase was a preliminary meeting with the users, to overcome potential difficulties), in order to minimise misinterpretations that could hinder the responses. Moreover, by ensuring anonymity and confidentiality, we hoped to ensure, at least partly, that participants provided more accurate feedback and genuine opinions, without the fear of being judged. Moreover, and as mentioned before, we tried to ensure reliable answers and feedback by setting different evaluation phases but maintaining a relatively stable group of people testing the application and rating its value and also the impact of the gamification elements; by doing so, we not only expected to ensure some continuity in the use of the application, and verify the impact of a regular use, but also to ensure this cross-validation, as a way of augmenting the reliance of the data gathered. Asking participants to provide their own comments and to mention some sustainable actions that they might have been implementing also helped fortifying the feedback received and avoid the tendency to just respond affirmatively without that corresponding to the truth (obviously, we recognise that these comments might also biased, but, at least, they are a step forward from the yes/no or agree/disagree options).

In future investigations, these limitations could be addressed or at least minimised by carrying out longitudinal studies to follow participants over time, allowing for a more in-depth analysis of trends and changes in variables of interest and usability of the application, as well of actual environmental impact, which could mitigate the problems raised by the self-reporting tools. Additionally, should the authors find it appropriate, the study could be replicated with larger sample sizes to increase statistical power and generalizability of findings, therefore building on this pilot testing and helping validate results obtained from the initial pilot study and provide more robust evidence to support conclusions drawn from the research.

## 5. Conclusions

In conclusion, the integration of gamification and web-based applications in sustainability education within the European context holds significant promise for enhancing the effectiveness of environmental awareness and action. This article explored the multifaceted benefits and challenges associated with the incorporation of these innovative approaches, shedding light on their potential to engage learners, foster behavioural change, and promote a culture of sustainability. The gamification of sustainability education offers a unique opportunity for engaging learners through immersive and interactive experiences. By incorporating game elements such as challenges, rewards, and competition, educators can captivate the attention of participants and encourage them to actively participate in their own learning journey. Moreover, the intrinsic motivation derived from gamified experiences can lead to prolonged engagement, making it a potent tool for instilling lasting behavioural changes towards more sustainable practices. The digital medium is also an advantage in itself, as it allows for real-time tracking of progress, enabling educators to adapt content and strategies to maximize learning outcomes. Furthermore, the scalability of web-based applications facilitates the dissemination of sustainability education across various European regions, transcending geographical limitations.

However, challenges exist in the successful implementation of these approaches. Designing effective gamified experiences requires a delicate balance between entertainment and educational content, avoiding potential pitfalls such as superficial engagement or

disengagement due to frustration. Likewise, ensuring equitable access to web-based applications is essential, addressing issues related to the digital divide and varying levels of technological literacy. Additionally, it is important to reckon that, in spite of the positive feedback received from the implementation of the application in Portugal, this study has its limitations, particularly the ones related to the relatively short number of people involved in this phase, which makes it difficult to generalise the positive results (even though nearly three other dozens of user have tested the application during the project's final event, these participants didn't answer the same questions as the ones in the pilot testing, which makes it difficult to compare results). Therefore, future tests and publications should engage more users, in order to allow us to paint a more throughout picture of the role of sustainability and pro-ecological applications in promoting long-term behaviour change and adoption of greener lifestyles.

That said, we still believe that the conducted research broadens the knowledge in the field of applications for the encouragement of pro-environmental behaviours, which has not been analysed extensively in the Portuguese context. The numbers gathered reinforce the potential of these tools to capacitate all citizens to participate in the ecological transition and the immensely positive feedback received, not only in the two-phase test presented here but also in other occasions, highlight the receptiveness of people to this kind of initiatives and corroborates what was presented in the beginning of the article—that many want to start acting more consciously but they don't hold the necessary information to do so. The GoBeEco project expected to raise awareness and increase knowledge to change behaviours, individual preferences, consumption habits and lifestyles of adult educators and learners in relation to ecology and climate protection. That is, the project hoped to develop educators' digital competence in education, so they are able to inspire learners to create behavioural changes in ecological habits and to educate them on the best possible ways to do so. Moreover, it aimed to provide them with the methodology to enable them to develop digital resources-based curricula based on using gamification or green skills and pro-eco attitudes. Thanks to the final analysis with end-users, project partners proved that, in sum, the project not only increased participants' eco-awareness but also inspired them to take tangible steps towards a more sustainable lifestyle.

At the very end of the project, a final evaluation took place in all countries involved in GoBeEco testify the value of the application and provide some clues and guidelines to the development of engaging and useful applications to foster pro-environmental behaviour:

Informative content and data: participants have stated an interest in having less theoretical information but more practical challenges; it seems to be consensual that daily tasks that gradually become habits are seen as a pathway to continuous learning.

- Real examples and statistics: providing real-world examples and statistics reinforces the impact of individual actions, empowering users with knowledge about their contribution to the environment.
- Reinforcement of the potential impactful change through daily habits: users express a heightened motivation when they perceive their actions as contributing to positive environmental change; in this sense, emphasizing the impact of altering daily habits resonates strongly with the audience.
- Concrete success indicators: the inclusion of small calculation examples illustrating $CO_2$ savings in task instructions is highly motivating. Users appreciate tangible evidence of their efforts, providing a sense of accomplishment and progress.
- Gamified and fun elements: Users find enjoyment in interacting with the app. Incorporating entertaining elements enhances user engagement and fosters a positive attitude towards habitual changes. On another hand, the inclusion of challenges, a badge system and user rankings add a competitive and rewarding dimension to the user experience, encouraging sustained participation.

The main elements discussed here, and its outcomes can be seen represented in the following concept map (Figure 6).

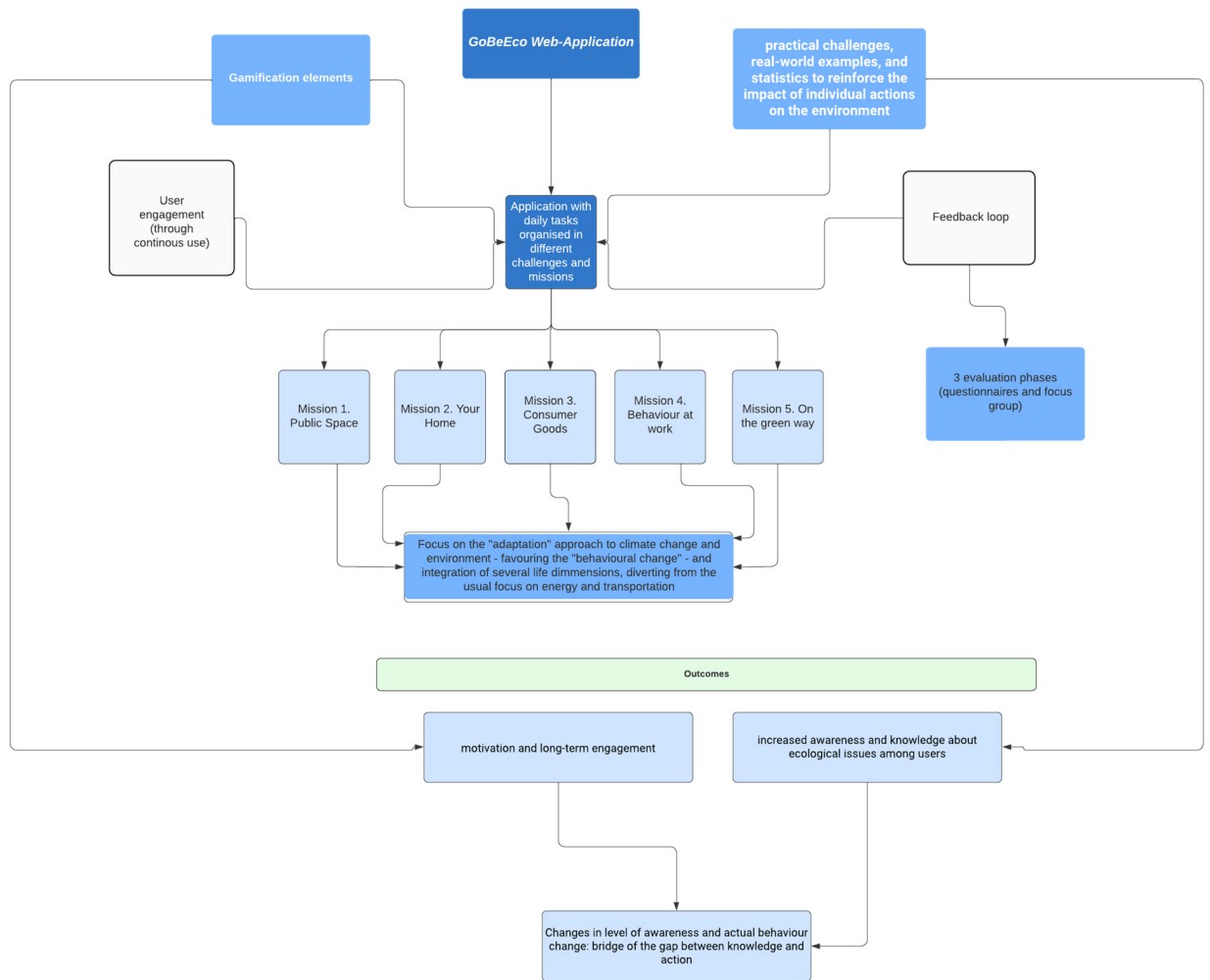

**Figure 6.** Concept map summarising the main contributions of GoBeEco.

In the future, to fully harness the potential of gamification and web-based applications in sustainability education, these aforementioned preferences, some of them corroborated also by other studies, but, additionally, the collaboration between educators, designers, researchers, policymakers, and relevant stakeholders is imperative [28]. By collectively addressing challenges and sharing best practices, the European educational landscape can evolve to offer innovative and impactful learning opportunities that contribute to a more sustainable future. The integration of gamification and web-based applications has the power to revolutionize sustainability education in the European context, fostering environmentally conscious citizens equipped to address complex global challenges. As we move forward, continued research, evaluation, and adaptation of these approaches will be essential to refine their effectiveness and ensure their alignment with evolving educational and societal needs. Closing this article, we would like to remind that all project's outputs are available online, completely accessible free of charge, and accessible in English, German, Polish, and Portuguese. The dedication to this free access resource allowed the partner to continue supporting it well beyond the project's initial lifespan. Moreover, there are expectations for the network to expand, as individuals who already benefit from the project's outcomes potentially assist newcomers in using those results. Consequently, we encourage experienced users to step up as volunteers within the network, providing frequent support to newcomers and helping, step by step, to change the world.

**Author Contributions:** Data curation, C.N. and C.Z.; Formal Analysis, C.N. and C.Z.; Methodology, C.V.d.C. and E.G.; Project Administration, E.G.; Supervision, E.G.; Writing—original draft, C.N. and C.Z.; Writing—Review and Editing, C.V.d.C. All authors have read and agreed to the published version of the manuscript.

**Funding:** This research was funded by the European Commission, grant number 2020-1-DE02-KA204-007517.

**Institutional Review Board Statement:** Ethical review and approval were waived for this study due to the following reasons: (a) the collected data was anonymized in such a way that the subjects cannot be traced back or their privacy infringed upon; (b) the research was designed for educational or instructional purposes and the included activities do not extend beyond normal educational or training practices in the involved organizations.

**Informed Consent Statement:** Informed consent was obtained from all subjects involved in the study.

**Data Availability Statement:** Data are contained within the article.

**Conflicts of Interest:** Authors Carolina Novo, Chiara Zanchetta were employed by the company Virtual Campus. The remaining authors declare that the research was conducted in the absence of any commercial or financial relationships that could be construed as a potential conflict of interest.

## Appendix A

*Appendix A.1. Questionnaire Used for the Evaluations Phases 1 and 2*

Dear participant,

This questionnaire aims to collect feedback on the GoBeEco gamification; in particular, to assess users' perceptions about the activities' content congruity, their experience with the app and their opinions on its usability and usefulness. Overall, we want to understand if GoBeEco is likely to have a positive impact on individuals' lives, and lead them to adopt more eco-friendly and sustainable practices.

Thank you for your contribution!

Participant's Information

1.  Country *
    - Germany
    - Poland
    - Portugal

2.  Partner organization (Please choose from the list below the organisation you have worked with/has contacted you to test the GoBeEco app.)
    - Fachhochschule des Mittelstands (FHM)
    - Energie Impuls OWL
    - PAIZ Konsulting
    - Ekopotencjal Foundation
    - Virtual Campus

    General questions

3.  I used the GoBeEco app via my smartphone *
    - Yes, via Android
    - Yes, via iOS
    - No

4.  I used the GoBeEco app (also) via a PC *
    - Yes, on Mozilla Firefox
    - Yes, on Opera
    - Yes, on Internet Explorer
    - Yes, on Google Chrome
    - Yes, on Safari
    - Yes, on Microsoft Edge

- No, I didn't use the app on a computer

5. At what time of day did you use the app the most? *
   - Morning
   - Midday
   - Evening

Usability and functionality

In relation to the usability and functionality of the app, we ask you to answer some questions pre- pared according to the System Usability Scale (SUS). When a SUS is used, participants are asked to score the following items with one of five responses that range from Strongly Disagree to Strongly Agree.

6. Please rate each question with one option ranging from Strongly Disagree (1) to Strongly Agree (5). *
   - I think that I would like to use this application frequently.
   - I found the application unnecessarily complex.
   - I thought the application was easy to use.
   - I found the various functions in this application were well integrated.
   - I thought there was too much inconsistency in this application.
   - I would imagine that most people would learn to use this application very quickly.
   - I found the application very cumbersome to use.
   - I felt very confident using the application.
   - I needed to learn a lot of things before I could get going with this application.
   - The app mechanisms (such as the structure of the mission challenge task or the way in which things are asked to be changed in everyday life) appealed to me very much.
   - The app components (such as points, badges, leaderboards, progress bar) are very well chosen.
   - The quiz at the end of the missions has once again consolidated my knowledge of sustainability in the subject area.

Impact on behavioral changes regarding ecology and pro- ecological awareness

Here we want to understand if GoBeEco contributed to increase your motivation to be more eco- friendly and to change your habits, and if you have interest in continuing to use the app.

7. Please rate each question with one option ranging from Strongly Disagree (1) to Strongly Agree (5). *
   - I have positively changed my behaviour in relation to sustainability by using the app.
   - By using the app, my awareness of ecological contexts has improved.
   - By using the app, I am motivated to engage more intensively with the topic of sustainability in my life.
   - I have been able to expand my knowledge of sustainability in the long term.

8. Comments/Suggestions

Thank you for your contribution

*Appendix A.2. Questionnaire Used for the Evaluation Phase 3*

Yes/No Questions and free text fields

1. Does the Gamification have the following features?

| | Yes | No |
|---|---|---|
| 1.1. Does the GoBeEco Gamification include dynamics (status, rewards)? | | |
| 1.2. Does the GoBeEco Gamification have different components (points, badges, achievements, leaderboards, notifications, levels, progress bar, tasks, challenges, missions)? | | |
| 1.3. Does the GoBeEco Gamification include different aesthetic aspects (e.g., narrative, challenge, discovery, sensation, fellowship)? | | |
| 1.4. Does the GoBeEco Gamification contain various behaviours (e.g., watch video, complete survey, answer quiz, read content)? | | |
| 1.5. Is GoBeEco Gamification a multisensory learning system? | | |

### 2.  What experiences did you have with the GoBeEco Gamification?

| | Yes | No |
|---|---|---|
| 2.1. The use of GoBeEco Gamification positively influenced my ecological behaviour. | | |
| 2.2. Please give 2-3 examples in what way! | Text | |
| 2.3. The GoBeEco Gamification raised my awareness for ecological contexts. | | |
| 2.4. By using the GoBeEco Gamification I am motivated to continue working on a more environmentally conscious behaviour. | | |
| 2.5. What motivated you the most? | Text | |
| 2.6. Do you have fun using the Gamification? | | |
| 2.7. In GoBeEco Gamification, appropriate areas (Missions/Challenges) help users improve their daily actions. | | |
| 2.8. In GoBeEco Gamification, clear habits are named and the app offers good guidance on how these habits can be made ecologically better. | | |
| 2.9. GoBeEco Gamification provides practical knowledge about what can be done for the climate in specific places I visit. | | |
| 2.10. What kind of practical knowledge was new for you? | Text | |
| 2.11. What was the biggest lesson on ecology you have learned with the Gamification? | Text | |
| 2.12. GoBeEco Gamification encourages different types and generation of users to take longterm action and develop an understanding of the importance of developing environmentally friendly habits. | | |
| 2.13. Do you feel now you could act as a change agent, to encourage others to develop themselves? | | |
| 2.14. Do you think it is worthwhile to go back to the gamification? | | |
| 2.15 Will you recommend the Gamification to others? | | |
| 2.16. Do you think, after using the Gamification, you indirectly inspire others to change habits? | | |
| 2.17. Do you have any comments from you or from other people using the app you can share here? Please fill in here the quotations: | Text | |

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
