# Peer review of "The Use of Gamification and Web-Based Apps for Sustainability Education"

_sustainability, doi:10.3390/su16083197_

Round 1

Reviewer 1 Report

Comments and Suggestions for Authors

The article in question presents the design, development and testing of an app to encourage adult learners to change their lifestyles to eco-friendly ones, focusing, in particular, on the implementation of the project in the Portuguese context (more specifically, the results of the pilot study carried out) . This app was developed by an international multidisciplinary team as part of a funded project.

The text addresses the relevance of using gamification and web-based apps for education for sustainability, presenting some supporting arguments and referring to some similar resources.

The resource development methodology is briefly presented, mentioning a participatory methodology in which, according to the authors, the collaboration of several partners was valued.

The text is well written and is, overall, clear in its structure and organization. The theme covered is current and relevant, being aligned with the commitment of the 2030 Agenda for Sustainable Development and the Sustainable Development Goals.

It is suggested, however, that the authors can make some improvements to the final version:

1 - The theoretical framework and the basis for the relevance of the study are aspects that should be further explored. The authors will be able to frame the relevance of the project and its results in the 2030 Agenda, presenting the resource as a contribution to the achievement of some of the SDG. The importance of education in implementing more sustainable actions can also be better substantiated.

2 - The presentation of other similar studies/apps can also be strengthened. There are several apps developed to promote sustainability education in formal, non-formal and even informal contexts.

3- In relation to the methodology followed, it would be important to clearly explain the purpose and research question at the beginning of the document. In relation to the assessment of the needs of potential users (carried out initially) and, subsequently, the evaluation of the app and its impact during piloting, it would be important to present the data collection instruments used.

4- It would also be interesting to understand which stakeholders were involved in the development of the app as well as the moment(s) and mode(s) of participation by end users.

Reviewer 2 Report

Comments and Suggestions for Authors

This article addresses the role of gamified digital tools in promoting self-awareness and reflection on daily habits and behaviours, guiding users towards consistent, long-term change towards a more sustainable lifestyle. Although the subject matter is intriguing, several gaps are identified that discourage its acceptance in a high-calibre journal such as this one.

Firstly, the abstract lacks crucial details, omitting the description of the method, results and conclusions. This lack of information is crucial for readers to understand the relevance and contributions of the study.

The introduction, on the other hand, is perceived as superficial, lacking sufficient references and failing to highlight the void that motivates the article, beyond the justification of the project itself.

Thirdly, the method is notoriously flawed. The lack of a detailed description of the participants and the limited information on the reliability and validity of the instruments used raise doubts about the robustness of the study. The results are considered insufficient, and the discussion and conclusions need substantial improvement to provide a deeper and more convincing analysis.

It is suggested that these shortcomings be addressed through a thorough revision of the article, incorporating essential details in the abstract, strengthening the introduction with solid references and clearly highlighting the gap that this study aims to address. It is also recommended to clarify and expand the method section, providing detailed information on the participants, reliability and validity of the instruments used. For the results, a more comprehensive analysis is urged, and significant improvement of the discussion and conclusions is encouraged for a more valuable contribution to the field.

Reviewer 3 Report

Comments and Suggestions for Authors

The use of a gamification strategy may contribute to a better awareness on the sustainability and ecological habits. This is the topic addressed on this paper. However, there are different aspects that should be revised on the paper.

1. The second paragraph of section 2 starts by mentioning about "respondents", however, it is not clear the number of participants in the initial inquiry neither the questions include in such inquiry.

2. Regarding the main study presented, there are 24 participants reported in the study, however, the results reported in table 2 have only 17 answers per question.

3. The instrument used is a self-reported instrument so the results may be biased. Moreover, the research sample is too low to obtain accurate conclusions about the ecological and sustainability habits.

4. It is not clear how the gamification contributes or is related to the ecological and sustainability habits. How are you evaluating the effectiveness of the gamification strategy applied?

5. Instruments should be described on section 2. In this paper, the instrument applied is described in section 3.

6. How did you come up with the empathy map presented in Figure 5? it is an authors' contribution or it is based on other author(s).

7. Discussion section should be broadened by commenting on more related studies.

Reviewer 4 Report

Comments and Suggestions for Authors

The research addresses the use of gamified web-based applications for supporting a more ecological and sustainable education and behaviour. In specific, it explores the GoBeEco EU-sponsored research project implementation in one of the corresponding consortium’s country partners, that is, Portugal.  This paper addresses gamification and web-based applications as L&D tools for sustainable education and behavioural change among adult cohorts. An original and enduring global need across academia, research, industry and communities.

It explores both an eco-gamification framework and actual implementation of this approach in GoBeEco application including learning challenges and user experience met by adult learners. We think that the authors should extend their result analysis and their subsequent discussion as their data reflect part of an expanded EU-research project.

We acknowledge the time and effort that authors spent in preparing and delivering their quality manuscript. To improve their submission further, we recommend the following steps, please:
a) Kindly expand a little bit the 1.1. The Digital world in favour of Sustainable Habits to include more gamification-related sustainability evidence/references;
b) Please extend your result analysis in more depth as your study seems to reflect a European research project performed;
c) Consequently, please expand on the current Discussion part;
d) Add a conceptual model that represents your final findings: this would support outcomes visualisation further;
e) cross-check with minor typos (e.g. l. 256: and not &; l. 163, 181, 208...application/s not app/s; l. 235: add a full stop).

The references are appropriate. The conclusions are consistent with the findings and arguments reported. However, we believe that they need to be extended to reflect the rather expanded nature of their EU-research context-specific project. Following the above recommended result extension, we anticipate that the tables and figures reflecting data expansion will be revised accordingly.

Comments on the Quality of English Language

The quality of English language is appropriate. 

Round 2

Reviewer 2 Report

Comments and Suggestions for Authors

Dear authors,

Great job! You have addressed all the improvement suggestions I provided regarding the abstract, introduction, methodology, and so on. The only remaining task would be to standardize and homogenize the format of the references, as it varies within the article despite them being documents of the same type. It would be beneficial for you to consult the journal's guidelines in this regard. Nevertheless, I want to congratulate you once again for the effort you have put in to enhance the overall quality of the article.

Best regards.

Author Response

Dear reviewer,

Thanks for all the work and contributions to the article. We have conducted an extended revision of the references and harmonized them according to the guidelines.